# Study on the Control of Membrane Fouling by Pulse Function Feed and CFD Simulation Verification

**DOI:** 10.3390/membranes12040362

**Published:** 2022-03-25

**Authors:** Chen Li, Dashuai Zhang, Jinrui Liu, Hanyu Xiong, Tianyi Sun, Xinghui Wu, Zaifeng Shi, Qiang Lin

**Affiliations:** 1Key Laboratory of Water Pollution Treatment and Resource Reuse of Hainan Province, Hainan Normal University, Haikou 571158, China; lichen_19930906@163.com (C.L.); jinruiliu1996@163.com (J.L.); 18162192028@163.com (H.X.); z793370994@163.com (Q.L.); 2School of Chemistry and Chemical Engineering, Hainan Normal University, Haikou 571158, China; 3School of Computer Science and Technology, Hainan Normal University, Haikou 571158, China

**Keywords:** membrane fouling, function feeding, shear force, CFD simulation

## Abstract

A complex-function fluid controller placed in front of a membrane module was used to control the velocity change with feed fluid and reduce membrane fouling. Using humic acid as the simulated pollutant, the effects of the square wave function, sine function, reciprocal function, and power function feeding on the membrane flux were investigated. For sine function feeding, the membrane-specific flux was the largest and was maintained above 0.85 under the intermittent frequency of 9 s. Compared with the final membrane-specific flux with steady-flow feeding of 0.55, functional feeding could significantly reduce membrane fouling. SEM results showed that sine feeding led to slight contamination on the membrane surface. Furthermore, the Computational Fluid Dynamics (CFD) simulation results showed that the shear force of sine function feeding was about three times that of the steady flow (6 × 10^5^ N). Compared with steady feeding, functional feeding could significantly improve the shear force on the membrane surface and reduce membrane fouling.

## 1. Introduction

In recent years, with industrial and domestic wastewater discharge, many water resources have been polluted by toxic organic compounds. The treatment of wastewater is an important global issue; it is imperative to find a suitable method to tackle the water contamination problem [1,2,3]. With the specific advantages of small installation size, low chemical consumption, easy maintenance, excellent separation efficiency, high water quality, and low environmental impact, ultrafiltration membrane technology has been widely applied in water purification [4]. However, the contaminant could plug the pores entirely through particle accumulation on the membrane surface, leading to an undesirable dramatic decline in permeate flux. Therefore, membrane fouling is a major challenge that limits the practical application of membranes. It is known that the key factors that control membrane fouling are the membrane properties, suspension composition, and hydrodynamic conditions [5,6]. There are many methods to reduce the concentration polarisation and migration in membrane fouling, such as feed solution pretreatment, chemical modification of the membrane, and optimization of the operational conditions [7]. Among these, feed solution pretreatment is the common method. Yang et al. [8] investigated the influence of UV/TiO_2_ photocatalysis as pretreatment to mitigate the cross-flow UF membrane fouling. With an extended UV/TiO_2_ pretreatment time, a significant reduction in the cake filtration coefficients and a slight decrease in the pore-blocking coefficients were observed. Tian et al. [9] applied ultraviolet/persulfate (UV/PS) pre-oxidation to control ultrafiltration membrane fouling. The results indicated that the NOM was considerably degraded and partially mineralized (~58%) by UV/PS pretreatment. However, the main disadvantage of membrane pretreatment is that it cannot fundamentally reduce membrane fouling, which limits the practical applications of this method.

Changing the operational conditions is considered an effective method to control membrane fouling. A large number of methods, including using a turbulence promoter, air sparging, and a different phase flow, have been used to increase the wall shear force and promote particle transportation from the membrane surface to reduce the agglomerate during the filtration process. To consider the effect of turbulence on hollow fiber membrane filtration in terms of membrane fouling performance, Pourbozorg [10] constructed a special setup with a turbulence generator using a vibrating perforated plate. The presence of turbulence moderated the membrane fouling and reduced the corresponding rate of transmembrane pressure (TMP) elevation. Wu et al. [11] studied a novel turbulence promoter based on the vibration gasket to control fouling. It achieved up to 48% and 25% more fouling mitigation than 1D (flat plate) and 2D (flat plate with grooves) spacers. Moreover, Liu et al. [12] simulated the turbulent flow in baffle-filled membrane tubes by CFD and further verified the results through experiments. The microfiltration of calcium carbonate suspensions was consistent with the CFD simulations, but the pressure drop along the baffle-filled membrane tubes was significantly increased owing to the frequent changes in the flow direction.

Previously, plenty of studies have pointed out that intermittent feeding, a significant method of controlling membrane fouling, is the key to further changing the velocity of liquid to produce flow turbulence. Taheri [13] proposed that intermittent relaxation is more effective for less compressible foulants than highly compressible and metastable foulants for mitigating fouling using a regenerated cellulose ultrafiltration membrane. They found that intermittent relaxation could mitigate humic acid fouling more effectively. Shi et al. [14] also found that pulse flow had a positive effect on membrane fouling control compared with steady flow. Rahmawati et al. [15] reported a novel finned spacer to form intermittent fluid to control membrane fouling. They found that panel permeability increased 22% compared with the conventional vertical system. All these approaches were used to control membrane fouling by manipulating the convective flow near the membrane surface and increasing the wall shear stress [16].

As a practical technology, CFD is a powerful tool for understanding the flow dynamics inside membrane modules by predicting the fluid flow and the heat and mass transfer through numerical simulation [17]. To reduce the cost, time, and risk associated with running repeated experiments, these processes are run using a numerical algorithm. The shear stress, an important parameter, should be investigated in the membrane module [18]. High-amplitude shear stress allowed for the reduction of the fouling layer thickness during the pulsatile flow [19]. Many experiments and CFD analysis studies have confirmed the relationship between shear stress and membrane performance [20]. Wu et al. [21] investigated various hollow fiber arrangements in a filter channel using CFD to improve membrane filtration operation. Tan et al. [22] also designed several types of 3D spacers and used CFD to simulate the shear force and turbulence along the membrane surface. Both simulation and experimental results revealed that the wave-like spacer could alleviate membrane fouling. This was because the high shear force on the membrane surface led to a remarkable influence on the back transportation of the contaminant, which could reduce the concentration polarization and the formation of the cake layer during the filtration process [23].

In this study, the complex function fluid controller was adapted to control the form of feed fluid that caused high shear force to reduce membrane fouling. The effects of intermittent flow on membrane flux, including the square wave function, sine function, reciprocal function, and power function feeding, were investigated. Scanning Electron Microscopy (SEM) was performed to reflect the cross-sectional contamination of the ultrafiltration membrane after different membrane fouling levels. In order to further understand the mechanism through which intermittent flow improves membrane flux, a CFD simulation was used to simulate and calculate the shear force during the UF process in FLUENT 18.0.

## 2. Materials and Methods

### 2.1. Materials

The following materials were obtained: sodium hydroxide (content ≥ 96.0%), humic acid (fulvic acid, FA ≥ 90%, Aladdin), a turbidity meter (2100AN, HACH, Loveland, CO, USA), a liquid flow timing controller (SFZ17--01, Haikou Changxiang Technology Co., Ltd., Haikou, China), a scanning electron microscope (JSM-7100F, JEOL, Tokyo, Japan), and a complex-function fluid controller (Haikou Changxiang Technology Co., Ltd., Haikou, China). The ultrafiltration membrane module (Hainan Litree Membrane Technology Co., Ltd. Haikou, China) with the product range 120 × 300 mm. The membrane was a hollow-fiber UF membrane made of polyvinyl chloride (PVC), and the molecular weight cut-off of the membrane was 20,000 Daltons.

### 2.2. Experimental Setup

Ultrafiltration membranes with a uniform pore distribution have been widely applied in drinking-water treatment. The membrane was a hollow-fiber UF membrane made of polyvinyl chloride (PVC), and the cut-off of the membrane was 20,000 Daltons. Figure 1 presents the experimental scheme of the process of cross-flow filter water treatment. The experiments were conducted using an automated filtration system consisting of a pump, flow meters, pressures, a membrane module, and a complex functional fluid controller. The functional fluid controller was program-controlled and could control the change in fluid shape with time as the square wave function, sine function, reciprocal function, and power function feeding by setting the form as an input function, which was applied before the membrane module to control the form of the feed fluid and reduce membrane fouling.

The normalized specific flux J/J_0_ was used to represent the membrane performance as a function of time, and J_0_ represented the initial membrane flux. For the UF experiments, the humic acid (HA) solution was diluted with deionized water in the stirred feed pool. The UF permeate and concentrate were run back to the feed tank.

### 2.3. Development of the CFD Simulation

CFD simulations were carried out on the different functional feeding processes to investigate the effect of shear force changes on the membrane permeation flux during the membrane separation [24]. The detailed simulation steps, model parameters, solution methods, and post-processing steps are discussed below.

#### 2.3.1. Governing Equations

The CFD simulation based on the three-dimensional module geometry was the numerical solution of the Navier–Stokes equations. In the present study, fluid was assumed as Newtonian, incompressible, isothermal, turbulent, and having constant physical properties under steady-state conditions. The fluid flow inside the module was simulated using the Realizable *k*-*ε* turbulence model. The time-averaged continuity and Navier–Stokes equations are as follows:(1)$$\frac{\partial \rho }{\partial t}+\nabla (\rho u)=0$$

Additionally, the N–S equations in the *x*, *y*, and *z*-directions are:(2)$$\frac{\partial (\rho u)}{\partial t}+div(\rho uu)=-\frac{\partial P}{\partial x}+div(\mu gredu)+S_{u}$$
(3)$$\frac{\partial (\rho \nu )}{\partial t}+div(\rho \nu u)=-\frac{\partial P}{\partial x}+div(\mu gred\nu )+S_{\nu }$$
(4)$$\frac{\partial (\rho w)}{\partial t}+div(\rho wu)=-\frac{\partial P}{\partial x}+div(\mu gredw)+S_{w}$$
where *P* is pressure (Pa), *μ* is dynamic viscosity, and ρ is fluid density (g cm^−3^). *S_u_*, *Sv*, and *S_w_* are generalized source terms of the momentum conservation equation.

In the simulation, the laminar flow model was used to simulate the hydrodynamics of the module under laminar flow operating conditions (Re < 2000), while the Realizable *k-ε* model was used for simulation under turbulent conditions caused by pulse flow (Re > 2000), where *k* is the turbulent flow energy (m^2^/s^2^) and *ε* is the turbulent dissipation rate (m^2^/s^3^). The transfer equation can be expressed as:(5)$$\frac{\partial (\rho k)}{\partial t}+\frac{\partial (\rho ku_{i})}{\partial x_{i}}=-\frac{\partial }{\partial x_{i}}\left[\left(\mu +\frac{\mu_{t}}{\sigma_{k}}\right)\frac{\partial k}{\partial x_{j}}\right]+G_{k}+\rho \varepsilon$$
(6)$$\frac{\partial (\rho \varepsilon )}{\partial t}+\frac{\partial (\rho \varepsilon u_{i})}{\partial x_{i}}=\frac{\partial }{\partial x_{j}}\left[\left(\mu +\frac{\mu_{t}}{\sigma_{\varepsilon }}\right)\frac{\partial \varepsilon }{\partial x_{j}}\right]+\rho C_{1}E\varepsilon +\rho C_{2}\frac{\varepsilon^{2}}{k+\sqrt{v\varepsilon }}$$
where
(7)$$\mu_{t}=\rho C_{\mu }\frac{k^{2}}{\varepsilon }$$
(8)$$\eta ={\left(2E_{ij}\cdot E_{ij}\right)}^{1/2}\frac{k}{\varepsilon }$$
(9)$$E_{ij}=\frac{1}{2}\left(\frac{\partial u_{i}}{\partial x_{j}}+\frac{\partial u_{j})}{\partial x_{i}}\right)$$
(10)$$C_{1}=\max\left(0.43,\frac{\eta }{\eta +5}\right)$$
(11)$$\sigma_{k}=1.0,\sigma_{\varepsilon }=1.2,C_{2}=1.9$$

#### 2.3.2. Model Geometry and Meshing

The fluid was assumed to be incompressible with constant fluid properties (viscosity and diffusivity). It was necessary to establish a model according to the actual situation and perform mesh division so that the governing equations were discrete in the spatial region, and the obtained discrete equations were solved. Reasonable meshing was essential to the numerical simulation results. In the module, numerous PVC membrane fibers were packed. However, the single-fiber membrane was considered in the simulation, which was because internal pressure feeding was used in the experiment. This means that the feed liquid flowed inside the membrane fiber so that the flow pattern of the liquid was considered the same for each membrane fiber. The computational domain included a cylindrical geometry with a diameter of 120 mm and a hollow fiber membrane with a diameter of 12 mm. The module body and hollow fibers were 300 mm in length. The inlet and outlet ports had a standard diameter of 12.5 mm.

The simulation model was established and divided by ICEM, the preprocessing software of FLUENT 18.0. In the module, several PVC membrane fibers were packed, but a single fiber was considered in the simulation. The membrane fiber was treated as the rigid element and divided into ten sections to monitor the change in the shear force near the membrane wall. The shear force near the membrane wall with different functional flow patterns was discussed. The fluid domain was discretized into 1,148,068 elements for the model. In order to ensure that the grid convergence index (GCI) was less than 5% in terms of fluid mechanics and mass transfer, a grid independent study could safely ignore potential sources of error in numerical calculations. The meshing geometry is shown in Figure 2.

#### 2.3.3. Boundary Condition and Solving Parameters

The boundary conditions were quite important for the numerical calculations. In the simulation, it was assumed that the fluid was incompressible and isothermal. Several different functional feed modes were investigated, including square wave functions, sine functions, reciprocal functions, and power functions. The inlet velocity of the function feeding was programmed by the user-defined function (UDF). The expressions of the four functional feeding velocities of inlet pores are shown in Table 1. The shear profiles at the inlet of the membrane module for the four functional feeds are shown in Figure 3.

The shear profiles arose from a change of velocity, which was controlled by a programmable complex function fluid controller. In both models, at the inlet port, the magnitude of velocity was determined as 1 m/s, and the direction of velocity was normal to the boundary. At the outlet port, constant gauge pressure was set to 10^5^ Pa.

#### 2.3.4. Numerical Methods

The shear force has been recognized as an essential parameter in controlling particle back-transport from the membrane surface. The functional feeding system was simulated via the first-order implicit method. The pressure–velocity coupling was handled via the SIMPLE algorithm, and the convective terms were discretized by a second-order upwind scheme. The mixture model and the standard k-ε model were used to describe the turbulent flow in the model, and the turbulent dissipation (ε) was introduced by the first-order upwind. The turbulence parameters were set by the hydraulic diameter and strength. As the default absolute convergence criteria, 10^−3^ was used for the solution variables [14].

## 3. Results and Discussion

### 3.1. Experimental Results

#### 3.1.1. Effects of Different Feeding Methods on Membrane Fouling Control

Several different functional feed modes were investigated, including sine functions, square wave functions, reciprocal functions, and power functions. The PVC ultrafiltration membrane was used to treat the sewage, which was simulated by humic acid, and all membranes were placed in distilled water for 24 h to remove impurities before use. A complex function controller to control the flow of liquid in the membrane in a functional manner was attached to the membrane inlet. The liquid film specific flux J/J_0_ and the plot with the time under different feeding methods are shown in Figure 4.

It was found that intermittent frequency had a great influence on membrane flux in previous experiments. The membrane-specific flux of different intermittent frequencies decreased with functional feeding. As seen in Figure 4a, the rapid decline in the early stage indicated that the feeding liquid entered the membrane module to produce the concentration polarization. Initially, the flux dropped sharply, and then the late decline trend gradually flattened, indicating that membrane fouling has occurred. It could be concluded that for square wave function feeding, the membrane-specific flux was the largest and could be kept at 0.8 when the duty time was 9 s and the running time was 4 s. For sine function feeding, the membrane-specific flux was the largest and was maintained above 0.85 when the intermittent frequency is 9 s. As for reciprocal function feeding, the membrane-specific flux decreased the slowest and was kept at 0.8 when the running and duty times were 8 s. For power function feeding, the membrane-specific flux decreased the slowest and was kept at 0.8 when the running and duty times were both 10 s. The excess duty time would cause the pollutant deposit on the membrane surface to accelerate the membrane fouling. However, if the duty time was too short, the particle size would become smaller, which could facilitate the blockage of the membrane pores and cause more serious pollution. Compared with the final membrane-specific flux of 0.55 of steady flow feeding, membrane fouling could be significantly reduced by functional feeding. Compared with the other functional feeding modes, the membrane-specific flux of the sine function decreased the slowest, and the membrane fouling was the lowest.

This is mainly because functional feeding could enhance the turbulence of the liquid and destroy the concentration polarization phenomenon on the membrane surface, improving the mass transfer efficiency. Furthermore, it could cause a certain degree of vibration on the membrane surface so that the deposit materials on the surface fell off. When the system suddenly stopped running, a pressure difference formed between the inside and outside of the module, which could cause the pollutants of the membrane surface to detach from the surface and be carried away by the water flow.

To comprehensively analyze the membrane-specific fluxes, the membrane maintained a high permeate flux when the operation and duty cycle times were 8–10 s for four functional feed modes. For sine function feeding, the membrane-specific flux was the largest and was maintained above 0.85 under the intermittent frequency of 9 s. A frequency that was too fast caused the particulate matter to become smaller and enter the membrane pores to block it, resulting in the membrane permeation flux decreasing. However, a frequency that was too slow caused the deposit sediments on the membrane surface to slowly compact to form a filter cake layer, which increased the membrane fouling. In summary, sine function feeding showed better performance in membrane fouling control.

#### 3.1.2. Membrane Surface Morphology Analysis

In order to determine the membrane surface contamination during the functional feeding process, the new and fouled membrane samples were assessed by scanning electron microscopy (SEM), which reflected the cross-sectional contamination of the ultrafiltration membrane after different membrane fouling levels. Figure 5 reflects the contamination of the membrane after different functional feeding treatments.

As seen in Figure 5a, the new membrane was composed of a support layer similar to a finger shape and a loose porous surface layer. The surface of the membrane was smooth and clean. The finger structure near the edge of the outer membrane might be responsible for the mechanical strength of the entire structure, which could reduce the movement of the liquid and prevent backflow during the cross-flow filtration process. Moreover, the strip profile on the membrane surface with the rich membrane pore structure was also very clear where no impurities were attached [24]. Figure 5b shows an electron micrograph of the membrane surface after periodic processing of the square wave function. Compared with the new membrane, its surface was slightly contaminated with fine particles but no filter cake layer. As for the membrane surface after the sinusoidal cycle treatment (Figure 5c), the membrane was slightly contaminated with the fine particle deposition, and there was no filter cake layer on the membrane surface. However, the decline in the actual operating flux might have been due to the deposition or adsorption of dirt on the membrane surface. Figure 5d-e show the SEM images of the membrane surface after the reciprocal function and the power function feeding, respectively. The filter cake layer had begun to form on the membrane surface. The SEM image of the membrane surface after the steady flow feeding treatment is shown in Figure 5f. The membrane surface was covered with a filter cake layer, and the membrane section with a large amount of granular material was rough. The formation of the filter cake layer affected the permeate and even prevented the filtrate from passing through the membrane pores. The pollution on the membrane surface was serious, but the deposition of humic acid particles was not observed. Because of the presence of pore-resistant contamination on the membrane, the average pore size of the contaminated membrane was significantly reduced, causing irreversible and reversible membrane fouling.

### 3.2. CFD Simulation of the Membrane Separation Process with Different Feeding Modes

#### 3.2.1. Effects of Feeding Modes on Shear Force

The shear force was proportional to speed, so the increase in the cross-flow rate could enhance shear force in the membrane process [25]. The shear forces near the membrane wall were distributed along the direction of fluid flow, as shown in Figure 6.

As seen in Figure 6, the shear force appeared to be the maximum at the membrane inlet, which was consistent with the previous study by Shi [14]. The results showed that the shear force increased with an increase in velocity. The presence of high velocity in the inlet ports could generate high shear stress on the hollow fibers and might have prevented cake formation on these fiber surfaces. The power function (*V* = *kt^a^* + *A* (*a* = 1, 2)) feeding could significantly increase the velocity of the feeding and further enhance the shear force near the membrane wall. The shear force reduction rates of the power function were 45.7% and 40.5% along the fluid flow direction, and the reduction rates of the sinusoidal function, square wave function, and reciprocal functional feeding were 28.6%, 38.7, and 45.4%, respectively. The shear force of the functional feeding was higher than that of the steady flow, while the increase in shear force in the membrane process could improve membrane fouling. The power function feeding had the largest shear force near the membrane wall, but the rate of decline along the flow direction of the liquid was also large. The sinusoidal feeding was most suitable for the long-term treatment of wastewater because it could cause relatively high shear force at the inlet and decreased the slowest along the flow of the liquid. In addition, the fluctuations of the inlet velocity not only caused the backflow of the feed liquid but also reduced the compression of the filter cake on the membrane surface or in the pores of the membrane, which could control membrane fouling.

#### 3.2.2. Effect of Feeding Modes on Membrane-Specific Flux

The experiments verified the simulation results of CFD. The membrane flux over time for functional feeding is shown in Figure 7.

The sinusoidal feeding maintained a high flux in one cycle, which was consistent with the simulation results. The pulsating flow under symmetric pressure could produce two equal maximum (bimodal) velocity distributions near the membrane wall surface rather than on the central line, thus improving mixing and reducing concentration polarization [26]. In Liang’s research using CFD simulation, the results showed that the sinusoidal slip velocity profiles could cause a similar increase in mass transfer and shear stress at a Reynolds number at which vortex shedding occurs [21]. From the point of view of hydraulic conditions, the sinusoidal functional feeding could significantly improve the membrane filtration process. Membrane fouling could be effectively controlled by regulating the shear force of the fluid on the membrane surface [27].

The results showed that membrane fouling could be effectively controlled by adjusting the fluid feeding mode and changing the shear force of the fluid on the membrane surface. The back-mixing force inside the feed liquid could be enhanced by increasing the turbulence of the membrane assembly feeding and the shear force on the membrane surface. The shear force on the membrane surface could be strengthened by increasing the degree of turbulence of the membrane module feeding [28]. It could be increased the back-mixing force inside the liquid and caused slow particle deposition on the membrane surface to increase the membrane flux and control the membrane fouling.

Notably, the power function had the largest shear force, but the actual membrane-specific flux was the smallest, which was inconsistent with the simulation results. This indicates that an overly large shearing force on the membrane surface could aggravate the phenomenon of membrane fouling. This is mainly because the excessive shear force could lead to particle breakage and particle screening, which can ultimately increase the membrane fouling resistance [29].

## 4. Conclusions

Using functional feed at the inlet of the membrane module could reduce membrane pollution. Functional feeding, such as the sine function, square wave function, reciprocity function, and power function, was used in the experiment, which could greatly improve the membrane flux. The membrane surface morphology was observed by SEM. In addition, the permeability of the membrane was tested and the shear force on the membrane surface was simulated by CFD. SEM showed that the fouling degree of membrane surface was different with different feeding methods. It is worth noting that the membrane of the sine function feeding treatment was slightly higher. In comparison with steady flow feeding, the membrane-specific flux of the functional feeding was significantly enhanced. Furthermore, the CFD simulation showed that the shearing force of sinusoidal feeding was about 3 times that of the steady flow (6 × 10^5^ N).

It was found that membrane fouling could be effectively controlled by adjusting the feeding mode of the fluid and changing the shear force of the fluid on the membrane surface. The turbulent deposition force on the membrane surface was controlled by changing the turbulent deposition force of membrane particles on the membrane surface, which thus increased the turbulent deposition force on the membrane surface. It should be noted that excessive shear forces could lead to particle breakage and particle sieving, which ultimately increased the fouling resistance of the membrane. In brief, the antifouling performance of functional feeding was better than that of steady feeding. Therefore, functional influent technology is worthy of further study and has broad application prospects in practical water treatment.

## Figures and Tables

**Figure 1 membranes-12-00362-f001:**
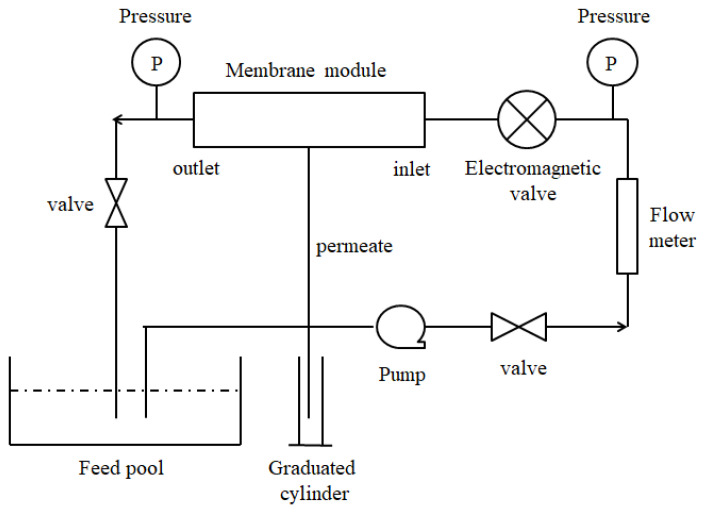
The cross-flow filter process diagram.

**Figure 2 membranes-12-00362-f002:**
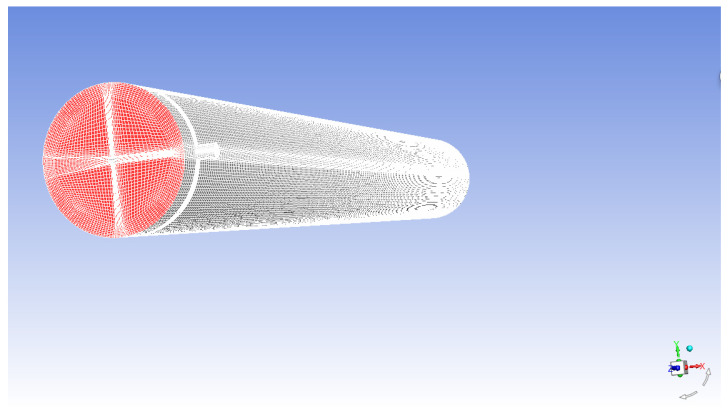
Membrane mesh geometry.

**Figure 3 membranes-12-00362-f003:**
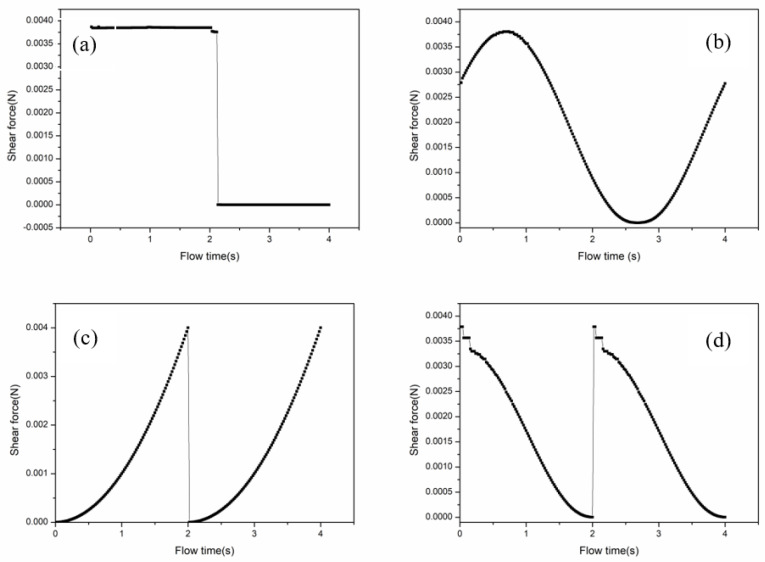
Shear profiles at the inlet of the membrane module for the four functional feed modes, (**a**) square wave, (**b**) sine, (**c**) reciprocal, and (**d**) power functional.

**Figure 4 membranes-12-00362-f004:**
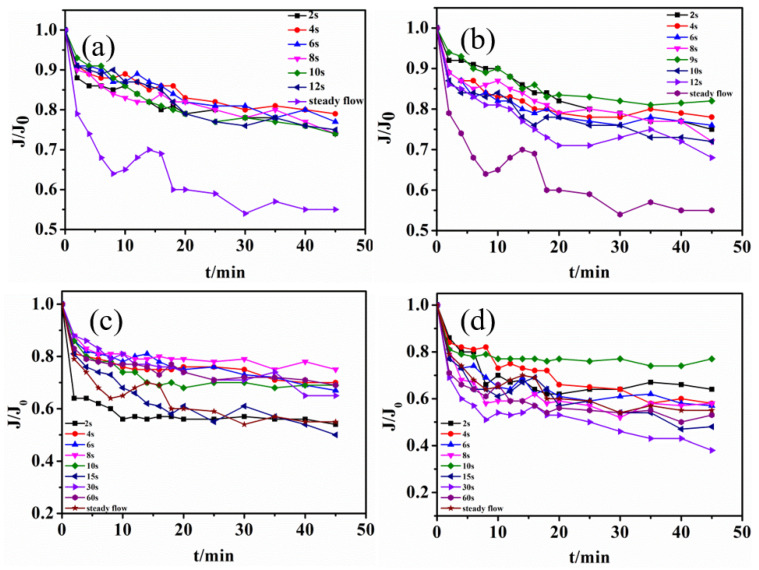
Membrane-specific flux–time diagrams of different function feeding modes (**a**) square wave, (**b**) sine, (**c**) reciprocal, and (**d**) power functions.

**Figure 5 membranes-12-00362-f005:**
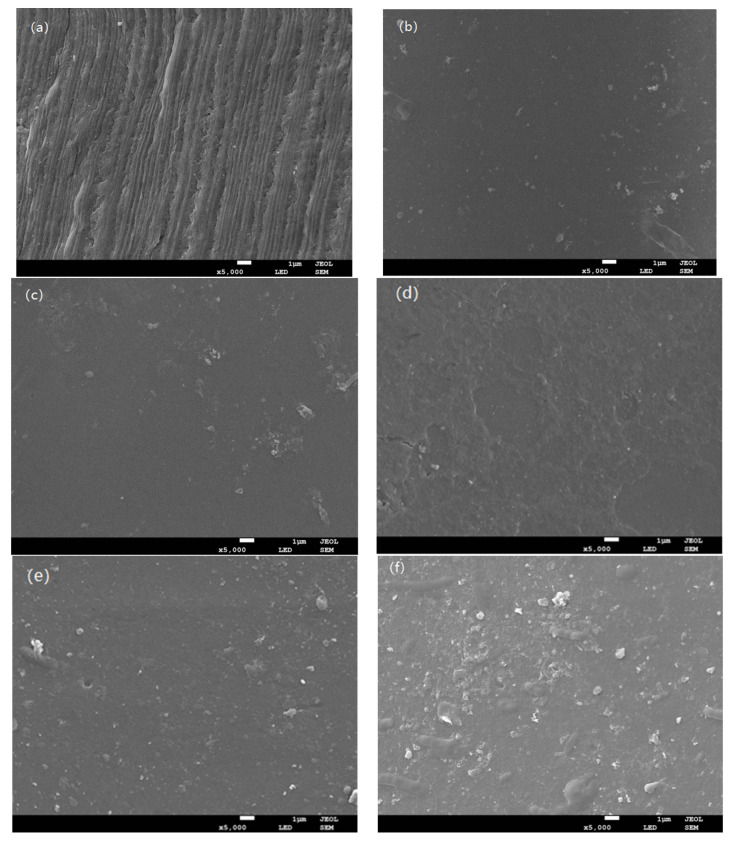
The SEM image of the surface of the membrane: (**a**) new membrane, (**b**) membrane surface with square wave function feeding, (**c**) membrane surface with sinusoidal function feeding, (**d**) membrane surface with reciprocal function feeding, (**e**) membrane surface with power function feeding, and (**f**) membrane surface with steady flow feeding.

**Figure 6 membranes-12-00362-f006:**
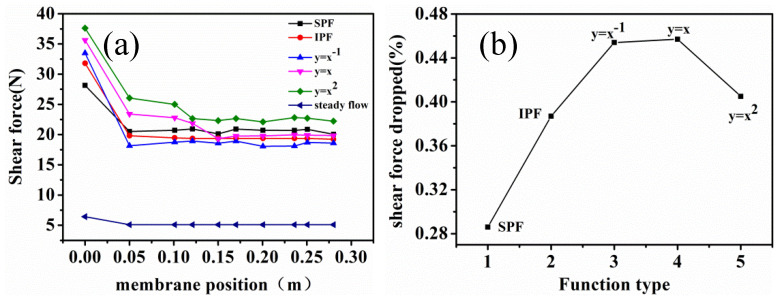
Comparison of the shear force of different function feeding modes, (**a**) shear force variation at different membrane positions, (**b**) the shear force dropped value.

**Figure 7 membranes-12-00362-f007:**
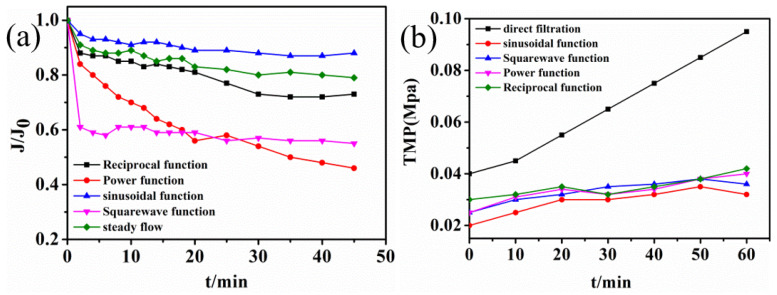
Different function feeding modes. (**a**) Membrane-specific flux time diagram. (**b**) Transmembrane pressure difference time diagram.

**Table 1 membranes-12-00362-t001:** Function expression of different feed modes.

Function Form	Function Expression
sine functions	$V=A+Asin\left(\frac{2\pi }{T}t+\varphi_{0}\right)$
square wave functions	$V=\{\begin{array}{c} 2A,\;\;x<t<\frac{T}{2}\\ 0,\;\;\frac{T}{2}<t<T \end{array}$
reciprocal functions	$V=k\;t^{a}+A\;\left(a=1,2\right)$
power functions	$V=k\;t^{-1}+A$

where *A* is pulse magnitude, (m/s), *T* is pulse period, (s), and *φ*_0_ is the initial phase.

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
