# Peer review of "Study on the Control of Membrane Fouling by Pulse Function Feed and CFD Simulation Verification"

_membranes, 2022, doi:10.3390/membranes12040362_

Round 1

Reviewer 1 Report

The manuscript entitled "Study on the Control of membrane fouling by Pulse function Feed and CFD Simulation Verification" presents an interesting route to solving the membrane fouling problem.  It is a meaningful topic in the field of membrane separation. However, there are several major comments that should be addressed to improve the quality of the manuscript and the journal:

  1. The figures in the manuscript should be improved, including the captions and labels. The current form is not well prepared. Improvement of English language and style is required. 
  2. The experimental section should be described more clearly for the potential readers repeating the work. Besides, the experiment results in figure 4 should have error bars indicating the fluctutation range. More detail description about the experiment results is needed to support the idea of membrane fouling control.  
  3. The abstract and the introduction should be improved. A clear backgroud description should added to the two sections, which could help the readers to better understanding of the main idea of this work. 
  4. This referee suggests the authors rewrite the conclution section which should be a deeper and further description of the main results of this work, rather than a repeating of the abstract section. 

Author Response

We are truly grateful to your and referees’ critical comments and thoughtful suggestions (including membranes-1628415-Research Paper .doc). Based on these, we have made careful modifications on the revised manuscript. All changes added to the text are in red. We hope that the revised manuscript will meet your Journal’s standard.

Reviewer 2 Report

The article has a suitable scientific content for publication, but the English must be improved since there are mistakes and expressions that do not sound proper along the main text body.

Author Response

(The authors gave the same response as above.)

Reviewer 3 Report

Review of the article entitled Study on the Control of membrane fouling by Pulse function Feed and CFD Simulation Verification

Authors: Chen Li, Dashuai Zhang, Jinrui Liu, Hanyu Xiong, Tianyi Sun, Xinghui Wu, Zaifeng Shi, Qiang Lin

In my opinion the subject of presented results is in agreement with the area of interest of the Membranes journal. The title is concise and informative and the manuscript is well organized. I find the paper suitable for publication in Membranes journal. However, I recommend the manuscript to be published after minor revision. The specific comments are presented below:

  1. Some editor errors should be improved.
  2. What does „migration the membrane fouling” mean? Line 40.
  3. Did the Authors determine the MWCO of membrane or the value was provided by manufacturer?
  4. What was the concentration of humic acids?
  5. CFD and other abbreviations should be defined at first mention.
  6. Table 1 has different resolutions.
  7. The figures, especially Fig. 4, 6, 7, have too low resolution to read them effectively and they are too small.
  8. The discussion with literature should be developed, especially in terms of proving Authors’ hypothesis presented in the 3.2.2 paragraph.

Author Response

Dear editor:

 We are truly grateful to your and referees’ critical comments and thoughtful suggestions (including membranes-1628415-Research Paper .doc). Based on these, we have made careful modifications on the revised manuscript. All changes added to the text are in red. We hope that the revised manuscript will meet your Journal’s standard.

  1. Some editor errors should be improved.

 These have been changed in the revised Manuscript.

  1. What does „migration the membrane fouling” mean? Line 40.

Line 40.“ migration the membrane fouling” have been changed “ migration in membrane fouling” in the revised Manuscript.

migration in membrane fouling(membrane fouling migration):In polluted membrane, the adsorption or deposition of particles, colloidal particles or solutes in the raw water on the membrane surface and membrane pores can cause the formation of gel layer caused by reversible concentration polarization. During the process of membrane filtration, physical or chemical action or mechanical action will cause the adsorption and deposition of contaminants on the membrane surface and membrane pores.

  1. Did the Authors determine the MWCO of membrane or the value was provided by manufacturer?

The MWCO of the membrane was provided by the manufacturer.

  1. What was the concentration of humic acids?

Preparation of humic acid simulated sewage: weigh 0.1g NaOH solid, dissolve it with ultrapure water, and transfer it to a 500ml volumetric flask for constant volume. Weigh 10 g of humic acid and dissolve it in NaOH solution, fully dissolve it and let it stand overnight. When in use, measure 50 ml of mother liquor and dilute it with deionized water to a turbidity of about 2 NTU. Humic acid concentration is about 0.6-0.7mg/L.

  1. CFD and other abbreviations should be defined at first mention.

       These have been changed in the revised Manuscript.

  1. Table 1 has different resolutions.

       These have been changed in the revised Manuscript.

  1. The figures, especially Fig. 4, 6, 7, have too low resolution to read them effectively and they are too small.

       These have been changed in the revised Manuscript.

  1. The discussion with literature should be developed, especially in terms of proving Authors’ hypothesis presented in the 3.2.2 paragraph.

       These have been changed in the revised Manuscript.

Round 2

Reviewer 1 Report

  1. The revised version of this manuscript has improved significantly. Most of the comments from this referee have been addressed.
  2. Improvement of English language and style is still required, sepecially the captions of figures. 

Author Response

Dear editor, the title of English language, style and number has been modified according to your requirements. Details of the modifications are in membranes-1628415 (2).
